# Discrimination and Characterization of *Escherichia coli* Originating from Clinical Cases of Femoral Head Necrosis in Broilers by MALDI-TOF Mass Spectrometry Confirms Great Heterogeneity of Isolates

**DOI:** 10.3390/microorganisms10071472

**Published:** 2022-07-20

**Authors:** Marina Nees, Michael Hess, Claudia Hess

**Affiliations:** Clinic for Poultry and Fish Medicine, Department for Farm Animals and Veterinary Public Health, University of Veterinary Medicine, Veterinaerplatz 1, 1210 Vienna, Austria; n.mary@web.de (M.N.); michael.hess@vetmeduni.ac.at (M.H.)

**Keywords:** *E. coli*, proteomics, standardization

## Abstract

*Escherichia coli*, a major pathogen in poultry production, is involved in femoral head necrosis (FHN) in broiler birds. So far, the characterization and relationship of isolates in context with this disease are mainly based on phenotypic and genotypic characteristics. Previously, an involvement of diverse *E. coli* isolates was reported. MALDI-TOF MS has been successfully applied investigating the clonality of different bacteria. Therefore, its application to characterize a well-defined selection of *E. coli* isolates beyond the species level was tested. The isolates were derived from clinical cases of FHN as well as from healthy birds. Reproducibility studies to perform a standardized protocol were done, and LB agar as well as the usage of fresh bacterial cultures proved most appropriate. No distinct clustering in context with the origin of isolates, association with lesions, serotype, or PFGE profile was found. Most of the isolates belonging to phylogroup B2 revealed a characteristic peak shift at 9716 *m*/*z* and could be attributed to the same MALDI-TOF MS cluster. The present study confirmed the previously found pheno- and genotypic heterogeneity of *E. coli* involved in FHN on the proteomic level. The study also highlights the need for standardized protocols when using MALDI-TOF MS for bacterial typing, especially beyond species level.

## 1. Introduction

Lameness or leg weakness is one of the most common and important issues in modern meat-type poultry production worldwide [1,2]. In association with pain and critically impaired motility, birds are unable to reach drinkers and feeders, which has to be seen as a welfare issue, with negative effects on the health and production parameters [3]. In most cases, the long bones of the hindlimbs are affected, with a focus on femoral heads. Here, gross pathological pictures range from degeneration characterized by separation of the cartilage up to necrosis and/or rupture of one or both femoral heads [4]. *Escherichia coli* is reported as one of the major agents causing femoral head necrosis—also known as bacterial chondronecrosis with osteomyelitis—in broilers [5]. So far, the characterization of *E. coli* isolates in context with disease outbreaks is mainly based on phenotypic characteristics like serotype and antimicrobial resistance profiles and genotypic characteristics like the presence/absence of virulence-associated genes or the determination of relatedness by phylo-grouping, PFGE or MLST [6]. In accordance with this, recent studies reported the presence of heterogeneous pheno- and genotypes of *E. coli* indicating the involvement of diverse isolates [4,6,7,8]. For microbiological identification but also characterization, proteomic phenotyping by matrix-assisted laser desorption/ionization time-of-flight mass spectrometry (MALDI-TOF MS) is widely used due to its broad applicability for Gram-positive and Gram-negative bacteria as well as for fungi [9]. It was shown that the accuracy of this method can be influenced by the culture conditions applied, e.g., media, incubation temperature, atmosphere and time, and storage parameters, with certain effects on the quality of spectra [10,11]. This is of special importance for investigations beyond species identification levels, as the validity of peak patterns which may lead to taxonomic markers can be negatively affected [12]. So far, MALDI-TOF MS has been successfully applied to investigate the clonality of different bacterial species like *Streptococcus pyogenes* [13], *Staphylococcus aureus* [14] and *Klebsiella pneumonia* [15]. In regard to *E. coli*, MALDI-TOF MS demonstrated a reliable determination of human relevant pathotypes [16] and a good discriminatory potential in regard to differentiate high-risk clones belonging to different phylogenetic groups [12]. Furthermore, it was shown that this method was able to correctly assign *E. coli* isolates derived from faecal contaminated surface water to specific source groups [17].

The aim of the present study was to apply MALDI-TOF MS in order to characterize *E. coli* isolates derived from clinical cases of femoral head necrosis in broilers as well as from healthy broilers. For this purpose, the first step comprised reproducibility studies using different growth media, storage conditions and age of cultured bacteria to implement a standardized protocol for processing the *E. coli* isolates. By determining their phylogenetic relationship, we wanted to reveal if isolates cluster based on their epidemiological data, namely the origin of isolate (flock, farm), if lesions—femoral head necrosis—are present, the serotype, the PFGE profile and the phylogenetic group.

## 2. Materials and Methods

### 2.1. Bacterial Strains

In total, 208 *E. coli* isolates from broilers were investigated. These strains were derived within a previous study from 21 farms, of which birds were investigated for the presence of femoral head necrosis at different ages and in which the *E. coli* isolates were pheno- and genotypically analysed [4]. Briefly, the isolates originated from 21 broiler farms. From farms 1, 2, 3, 4, 6, 7, 12 and 16, all samples were derived from a single flock. From the remaining farms the samples originated from two flocks, except for farm 21, where samples from 3 flocks were included in the present study. The isolation of *E. coli* was performed from femoral bone marrow by aseptically cutting left and/or right femoral heads in half. The material of the bone marrow was streaked directly on MacConkey agar (Scharlau, Vienna, Austria). The agar plates were incubated aerobically at 37 °C for 24 h. A total of 107 isolates were derived from birds at the age of one to two weeks, and 101 isolates were derived from birds at four to five weeks of age. Labeling of isolates was done based on year of isolation/running diagnostic number-number of bird with isolation site (KnoLi: bone marrow of left femur; KnoRe: bone marrow of right femur) and clone number. 148 isolates originated from birds with femoral head necrosis, and 60 isolates originated from not affected femora. The isolates were classified based on their lesions, serotype, PFGE profile and phylogroup (Appendix A). All isolates were stored at −80 °C by adding 2 mL of 40% glycerol/10 mL Brain Heart Infusion Broth (Oxoid, ThermoFisher Scientific, Vienna, Austria).

### 2.2. MALDI-TOF MS

#### 2.2.1. Reproducibility, Influence of Media and Age of Bacterial Culture

Elucidating the most suitable solid media to perform MALDI-TOF MS analysis on *E. coli* isolates beyond species level, four different agars were tested: blood agar (Columbia agar supplemented with 5% sheep blood, BioMeriéux, Vienna, Austria), MacConkey agar (Neogen, LabM, Heywood, UK), Chromocult^®^ Coliform agar (Merck KGaA, Darmstadt, Germany) and Luria Broth (LB) agar (Oxoid, ThermoFisher Scientific, Vienna, Austria). The consistency and reproducibility of spectra were used as parameters with three *E. coli* isolates each being selected from farms 1, 15, 20 and 21, resulting in a total number of 12 isolates for the test setting. These isolates displayed an identical serotype and phylogenetic group based on their farm origin. The same identical patterns were also found with regard to the PFGE profile, except for one isolate from farm 15 (Table 1). Independent of the agar, isolates were thawed for cultivation, and cultures were incubated at 37 °C for 24 h under aerobic conditions.

During this investigation, we found that the isolates from farms 15 and 20 exhibited specific peaks: isolates from farm 15 at 7400 *m*/*z* and 7560 *m*/*z*, and those from farm 20 at 6725 *m*/*z* and 6860 *m*/*z* (Table 1). Therefore, these isolates were selected to investigate the influence of sub-culturing, which was performed for three consecutive days. Besides this, these isolates were also used to test the influence of the age of bacterial cultures by storing them for 24 h, 48 h and 72 h at fridge temperature (4 °C ± 1.5 °C). For both test settings, each culture was analysed daily by MALDI-TOF MS. The spectra gained were compared to the spectra from fresh cultures received after thawing and incubation for 24 h at 37 °C.

To test the technical reproducibility of the spectra, the quality obtained for each strain was measured three times.

Finally, after implementing a standardized protocol for processing the *E. coli* isolates, all 208 strains were investigated.

#### 2.2.2. Sample Preparation, Proteomic Phenotyping

Sample preparation as well as parameters and data visualization and analysis using a Microflex LT instrument (Bruker Daltonics GmbH, Bremen, Germany) were performed as previously described [18], including minor modifications. Briefly, sample preparation for MALDI-TOF MS analysis was performed according to the standard protocol from the manufacturer (Bruker Daltonics GmbH, Bremen, Germany). After incubation at 37 °C for 24 h, two or three colonies were removed from the LB agar plate using an inoculating loop. The bacteria were suspended in 300 μL ultra purified water in a 1.5 mL Eppendorf tube. To inactivate the bacteria, 900 μL absolute ethanol was added to the suspension. After centrifugation for 2 min at 13,000× *g* the supernatant was discarded. Subsequently, the residual ethanol was removed with a second centrifugation step. To disrupt the cell wall, 50 μL formic acid (70%) was added to the pellet and mixed. Afterwards, 50 μL acetonitrile was added for protein extraction. After the last centrifugation step at 13,000× *g* for 2 min, 1 μL supernatant was spotted on the steel MALDI target plate and left to dry at room temperature. For database construction, each sample was spotted eight times on the MALDI target plate and each spot was measured three times to get 24 spectra for each strain. The sample spots were overlaid with 1μL matrix (α-cyano-4-hydroxycinnamic acid in 50% acetonitrile/47.5% water/2.5% trifluoroacetic acid) and dried. All steps were performed at room temperature. For calibration and optimization of the MALDI-TOF MS measurements and the efficiency control for sample identification, BTS (Bruker Bacterial Test Standard) was included in each measurement. The parameter settings for the Microflex LT instrument were as follows: IS1, 20.00 kV; IS2, 16.60 kV; lens, 7.00 kV; detector gain, 2974 V.

Two hundred and forty laser shots in 40 shot steps (in the linear, positive ion mode with 60 Hz nitrogen laser from different positions of the target spot) were summarized automatically with the AutoXecute (MBT AutoX method) acquisition control software (Flex control 4; Bruker Daltonics). For automated data analysis, raw spectra were processed using MALDI Biotyper software (Bruker Daltonics) with the default settings. The software performs smoothing, normalization, baseline subtraction, and peak picking, thereby creating a list of the most significant peaks (*m*/*z* values) of the spectrum. For species identification, the MALDI Biotyper output is a log (score) in the range of 0.00 to 3.00, computed by comparison of the peak list for an unknown isolate with the reference Main Spectra (MSP) in the reference database. A MALDI score between 2.00 and 3.00 represents identification on a species level, a MALDI score between 1.70 and 1.99 represents identification at genus evel, and anything less 1.69 is given as non-identifiable by the software. For the construction of the *E. coli* database and MSP creation used for the actual investigations, from each strain acid-soluble proteins were spotted on the MALDI target plate eight times. Each spot was measured tree times resulting in 24 spectra for each strain. FlexAnalysis (v.4) software (Bruker Daltonics) was used or visual inspection of the mass spectra and atypical spectra were excluded from further analysis (e.g., flat line spectra, spectra containing high matrix background signals). Twenty to 24 mass spectra were processed for each strain. The spectral peak lists were transferred into MSPs containing information on average peak masses, average peak intensities and peak frequencies. Similar MSPs result in a high matching score value. Each MSP was compared with all MSPs of the analyzed set. The list of score values was used to calculate normalized distance values between the analyzed strains resulting in a matrix of matching scores. The visualization of the respective relationship between mass spectrum profiles (MSP) is displayed in a dendrogram using the following settings of the MALDI Biotyper 4.1 software: distance measure was set to correlation, linkage to average and score threshold value for a single organism at 700. Based on Sauer et al. [19], clusters of strains with distance levels < 500 were classified as species. For strain relationship visualization, a dendrogram was formed based on MSP.

## 3. Results

### 3.1. Reproducibility, Influence of Media and Storage Conditions

All 12 isolates used for reproducibility testing were correctly identified as *E. coli* by MALDI-TOF MS independent of the solid media used with log(score) values above >2.0. Despite this, a clear negative influence on the quality of spectra in regard to their intensity and consistency was found when isolates were cultured on blood, MacConkey and Chromocult^®^ Coliform agar characterized by the low-resolution of spectra with overlapping peaks. Additionally, in the case of MacConkey agar, some proteins were not expressed, resulting in less peaks present in the spectra. Based on these findings, blood, MacConkey and Chromocult^®^ Coliform agar proved unsuitable for characterization of *E. coli* beyond species level and only cultures grown on LB agar provided reproducible and good-quality spectra (Figure 1). Therefore, this agar was subsequently used for all further investigations.

The negative influence of culture refreshment on the quality of spectra by preparing sub-cultures was demonstrated on six selected *E. coli* isolates. We were able to show that with every sub-cultivation step the intensity of characteristic peaks decreased. Additionally, peaks in the range 2000 to 6000 *m*/*z* were also affected displaying reduced quality (Figure 2).

When storing fresh *E. coli* cultures at fridge temperature up to 24 h, no differences in the quality of spectra were found. But a clear decrease of spectra quality was found when cultures were stored >24 h. Interestingly, an additional peak at 6535 *m*/*z* was found when cultures were stored at 4 °C independent of the time period (Figure 3). Based on this, only fresh cultures without any storing period were used for further investigations.

### 3.2. Proteomic Phenotyping

Using the above mentioned MALDI database and the standardized protocol for culturing the 208 *E. coli* isolates, all were identified to species level by MALDI-TOF MS with log(score) values above > 2.0. The score oriented MSP dendrogram of all isolates revealed four distinct clusters at the arbitrary distance level of 500. Cluster 1 comprised the lowest number of isolates, containing only seven strains. Seventy isolates were allocated into cluster 2, and 26 were allocated into cluster 3. The majority of isolates (*n* = 105) were grouped into cluster 4 (Appendix A; Figure 4).

No distinct clustering was noticed in context with the origin of isolates. Independent from the farm or flock, the isolates were distributed in all four clusters. Similarly, isolates did not group based on the appearance of clinical lesion. Isolates derived from femoral head necrosis shared the same clusters with isolates from healthy birds. It was also not possible to detect a certain arrangement of isolates based on their serotypes as isolates grouped independently from this phenotypic characteristic. In the same way, MALDI-TOF MS did not reveal a grouping of *E. coli* isolates based on their PFGE profiles. Furthermore, when setting the cut-off at distance levels lower than 500, we were not able to determine phylogenetic relationships or clonal lineages between isolates based on the mentioned characteristics.

Interestingly, the correlation between phylogenetic groups and MALDI-TOF MS was found as a result of a peak shift distinctive for the majority of isolates (47/53) belonging to the phylogenetic group B2 at 9716 *m*/*z*. Additionally, we also found six isolates from this group, revealing a peak at 9741 *m*/*z*. This peak was shared by all isolates belonging to phylogroups A, B1 and D. However, all *E. coli* isolates with the characteristic peak at 9741 *m*/*z* were distributed in different clusters. In Figure 5A–E we show the clustering of 50 isolates randomly selected from eight farms based on their origin, the presence of femoral head necrosis, their serotypes, their PFGE profiles and their phylogrouping.

## 4. Discussion

Previous studies have shown that lameness in broilers due to femoral head necrosis is very often related to *E. coli* infections [5,8,20]. However, in contrast to the assumption that certain types of avian pathogenic *E. coli* might be predominantly responsible for the disease pheno- and genotypic studies proved a great heterogeneity among these isolates [5,8]. In recent years, proteomics became popular not only in regard to bacterial identification but also for the characterization beyond species level and the determination of isolates relatedness. With this, the method also became more attractive to veterinary diagnostic laboratories, facilitating better knowledge of epidemiological data. Therefore, the present study intended to investigate the discrimination potential of MALDI-TOF MS to characterize a selection of 208 well-defined *E. coli* isolates originating from a previous study on femoral head necrosis in broilers [4]. For this, the application of a standardized sample preparation protocol is presupposed. Former studies clearly showed that the media used for growing the bacteria may alter the quality of spectra obtained from MALDI analysis [10,21,22]. Therefore, in a first step, the applicability of different media for MALDI-TOF MS analysis of *E. coli* was investigated. We were able to show that Columbia blood agar clearly decreased the quality of protein spectral profiles obtained for *E. coli* in a similar way as was previously shown for *S. aureus* [23]. The authors of that study were able to show that additional peaks were derived from breakdown products of the blood, hindering identification and further characterization. Furthermore, the use of MacConkey agar was reported to have a negative effect on MALDI results for a variety of bacteria [24,25]. This is in agreement with our findings, and was also seen when culturing *E. coli* on Chromocult^®^ Coliform agar. A possible explanation for the actual poor quality of spectra obtained in our study when using these media might be due to the inhibitory effects caused by the diverse agar components leading to metabolic influences on the bacteria. This may result in a decreased amount of ribosomal proteins which leads to less intense proteomic peaks. In contrast, LB agar contains only a few components, and proved superior for the analysis of *E. coli* in gaining reproducible and high quality spectra. This finding is in agreement with previous studies on different bacteria [26]. The quality of spectra was also negatively influenced by the age of the bacterial culture, by sub-cultivation and by the storage conditions, a finding also reported for other bacteria [11,27,28]. However, not only the quality of spectra was affected by the storage conditions, as an alteration of spectra by the appearance of an additional peak was found when isolates were stored prior to their investigation at 4 °C. As no analysis of proteins was performed, the nature of this additional peak remains unclear. However, stresses such as cold are known to induce a variety of bacterial adaptive techniques, like stress protein production, which have an influence on the phenotypic and genotypic level as shown for *E. coli* [29]. The reported findings are of general interest and highlight the importance of using a standardized protocol when applying MALDI-TOF MS for relationship determination of isolates or comparing outcomes from different studies. We were able to confirm previous data that the growth medium does not primarily disturb identification on species level but strongly affects the potential to differentiate strains beyond that level [30].

Previous studies demonstrated the applicability of MALDI-TOF MS to define clonal lineages of bacterial isolates according to their epidemiological origin [17,18,31]. This could not be confirmed for *E. coli* isolates investigated in the actual study. A phylogenetic analysis based on MSPs dendrograms revealed the origin-independent allocation of isolates in four clusters, confirming the assumption that broiler chicken harbor a very heterogeneous population of *E. coli* which are also involved in femoral head necrosis [4]. The determination of *E. coli* serotypes is widely used to confirm the presence of avian pathogenic *E. coli* (APEC). Serotypes O1:K1, O2:K1 and O78:K80 are predominantly attributed as the cause of diseases in poultry [32]. Therefore, we investigated whether MALDI-TOF MS mass spectra of *E. coli* cluster with serotyping. In agreement with studies on *S. pneumonia* and *O. rhinotracheale*, no reliable correlation was found [33,34]. Ojima-Kato et al. [35] reported the successful discrimination of certain enterohemorrhagic *E. coli* from other *E. coli* serovars, which is in contrast to our findings. Approximately half of the isolates in our study could be assigned clinically as APEC, but MALDI-TOF MS analysis did not reveal a phylogenetic difference to non-APEC isolates. PFGE is widely used for the typing of bacteria [36]. However, similar to previous studies, we were not able to find a clustering of identical PFGE profiles from *E. coli* when applying MALDI-TOF MS [12,37]. It has to mentioned that no other computational methods were applied in the present study which might have an effect on the formation of strain clusters, as shown for S. aureus [38].

A good discriminatory potential of MALDI-TOF MS was previously reported with regard to the *E. coli* phylogenetic group B2 from those of groups A, B1 and D [12,39]. In agreement with this, a distinct peak shift at 9716 *m*/*z* was found in the majority of the actual B2 isolates, and all isolates belonging to phylo-groups A, B1 and D revealed a known peak shift at 9741 *m*/*z*. However, we also detected B2 isolates which exhibited a peak shift at 9741 *m*/*z*, a feature previously unreported. So far, investigations on phylogroups with MALDI-TOF MS were only performed on human isolates. Consequently, our study is the first investigating *E. coli* phylogroups from broiler chickens. Based on our findings, it can be hypothesized that B2 isolates from broilers may comprise a wider range of MALDI types compared to humans. It would be interesting to confirm this finding in additional studies with different clinical backgrounds, keeping in mind that the use of different growth and sample preparation protocols, different instrumental settings and even instrumental variations can cause differences in gained mass spectra of the same bacteria, hindering the comparison of independent data sets [40,41].

The present study confirmed the previously found pheno- and genotypic heterogeneity of *E. coli* isolated from broiler chickens on a proteomic level. The study highlights the need to set up standardized protocols when using MALDI-TOF MS for bacterial typing beyond the species level. By this the improvement of spectra quality and reliable and reproducible results can be assured and robust relationships can be determined.

## 5. Conclusions

In the present study we demonstrated the importance of applying standardized protocols for *E. coli* typing by MALDI-TOF MS beyond the species level. By including several parameters in reproducibility tests we were able to show the influence on the quality and consistency of spectra by (i) the agar media, (ii) the influence of sub-cultivation, and (iii) different storage conditions. We were able to confirm a certain discriminatory power of MALDI-TOF MS in regard to the *E. coli* phylogroup B2 in broiler isolates, revealing a peak shift at 9716 *m*/*z*, known from human isolates. However, we also present data on phylogroup B2 isolates which exhibit the same peak shift as shown for A, B1, and D at 9741 *m*/*z*, a feature which needs to be investigated in detail in future studies. Investigating the relationship of 208 well-defined *E. coli* isolates a lack to differentiate isolates based on origin, lesions, serotypes or PFGE types was revealed, questioning the assignment of strains as APEC based on MALDI-TOF MS.

## Figures and Tables

**Figure 1 microorganisms-10-01472-f001:**
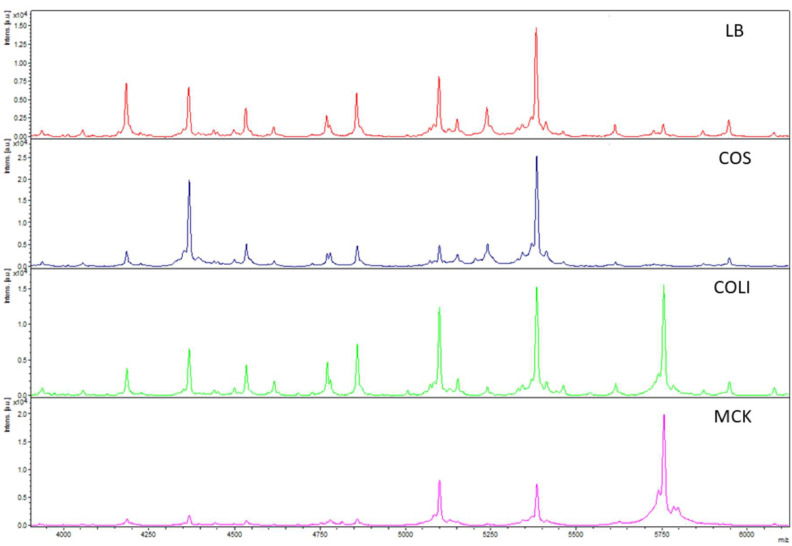
Reproducibility of spectra on different solid agars shown for *E. coli* isolate 16/01363-1 KnoLi 2 (LB: Luria Broth agar; COS: Columbia agar with sheep blood; COLI: Coliformen agar; MCK: MacConkey agar). Differences in intensity and quantity of peaks are shown within the range of 4000–6000 *m*/*z*.

**Figure 2 microorganisms-10-01472-f002:**
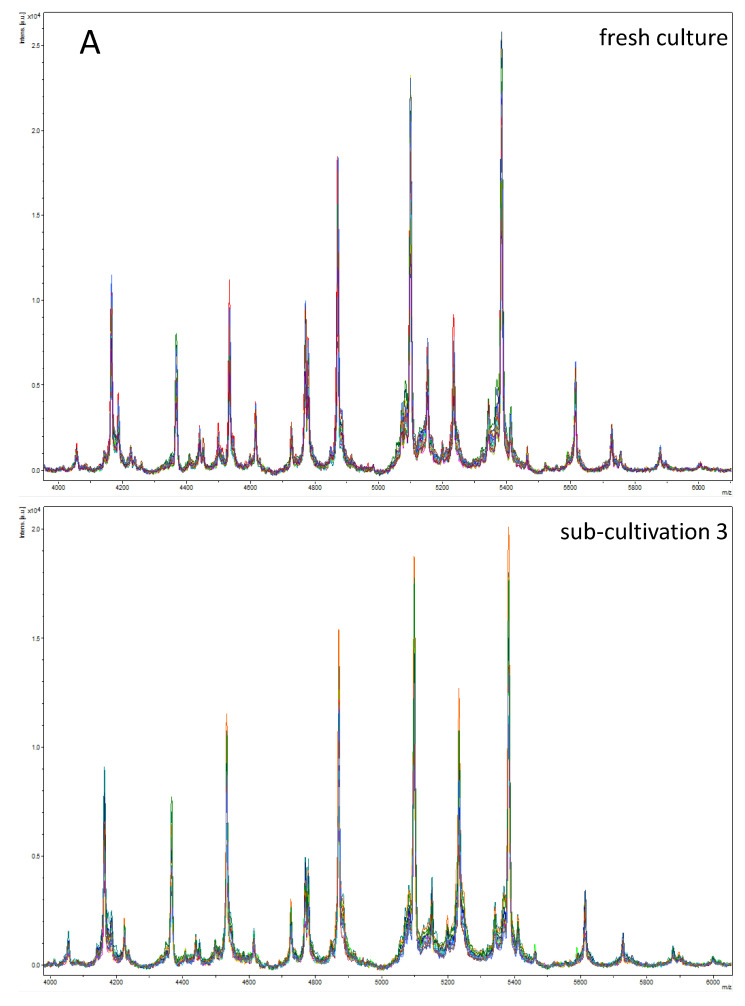
Influence of sub-cultivation (fresh culture vs. sub-cultivation step 3) on spectra quality with a decrease of peak intensity. Shown for *E. coli* isolate 16/1523-3 KnoRe 2 (*n* = 20 spectra). (**A**) Within the range of 4000–6000 *m*/*z*; (**B**) Within the range of 6000–8000 *m*/*z*; (**C**) Within the range of 8000–10000 *m*/*z*.

**Figure 3 microorganisms-10-01472-f003:**
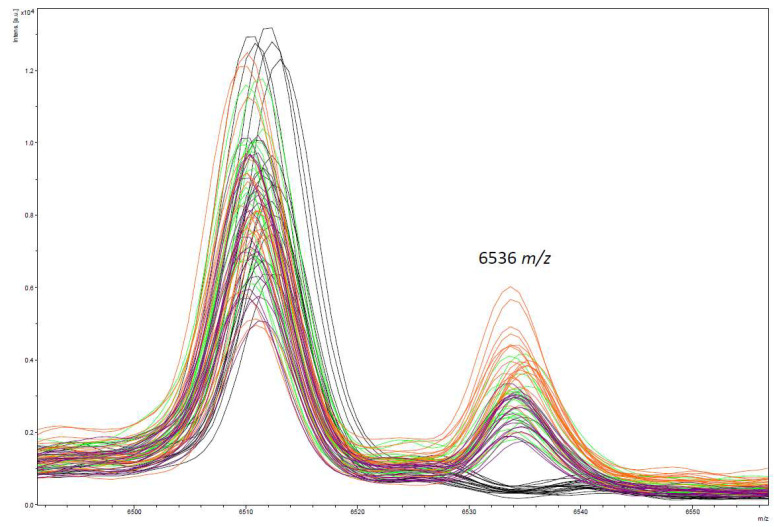
Additional peak (6535 *m*/*z*) detected in all *E. coli* strains when stored at 4 °C (24 h: green lines, 48 h: orange lines, 72 h: violet lines) compared to the fresh culture (black).

**Figure 4 microorganisms-10-01472-f004:**
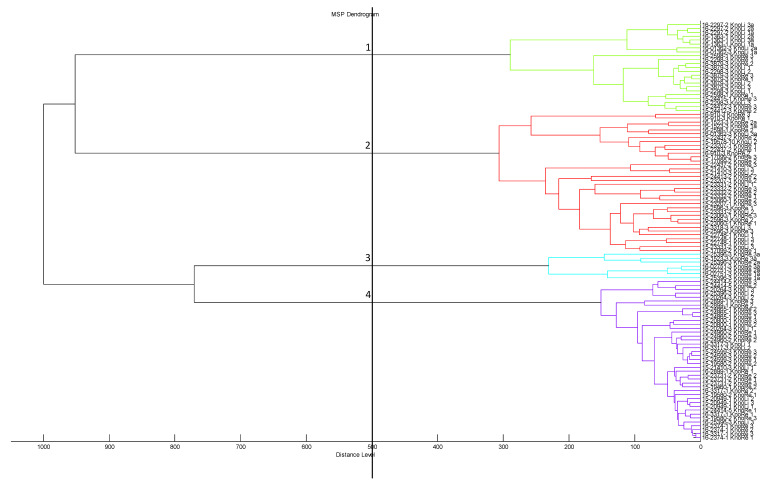
Score oriented MSP dendrogram based on 107 *E. coli* isolates revealing four distinct clusters (distance level 500).

**Figure 5 microorganisms-10-01472-f005:**
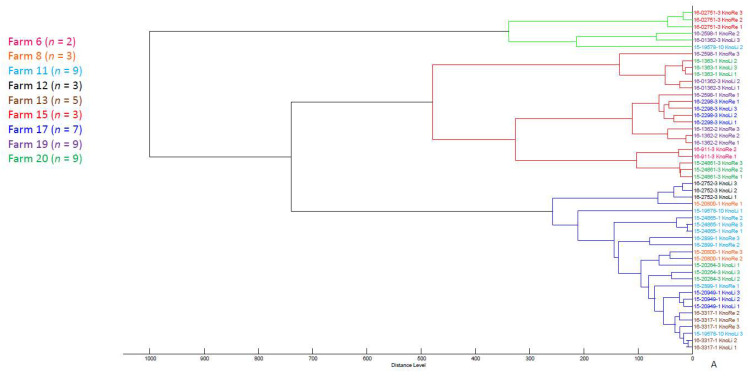
Clustering of 50 randomly selected *E. coli* isolates from eight farms based on their farm origin (**A**), the presence of femoral head necrosis (**B**), their serotype (**C**), their PFGE profile (**D**), and their phylogroup (**E**).

**Table 1 microorganisms-10-01472-t001:** *E. coli* isolates selected for standardization process of the protocol, based on their farm origin, serotype, PFGE profile and phylogroup. The presence/absence of specific peaks is given on which isolates from farms 15 and 20 were selected for testing the influence of sub-culturing, age of bacteria and storage condition.

Farm	Isolate	Serotype	PFGE Profile	Phylogroup	Specific MALDI Peaks
1	16/02297-2 KnoLi 1	O2:K1	BR14	B2	not detected
	16/02297-2 KnoLi 2	O2:K1	BR14	B2	
	16/02297-2 KnoLi 3	O2:K1	BR14	B2	
15	16/01523-3 KnoRe 1	neg	BR8	D	7400 *m*/*z* and 7560 *m*/*z*
	16/01523-3 KnoRe 2	neg	BR8	D	
	16/01523-3 KnoRe 3	neg	BR1ST1	D	
20	16/01363-1 KnoLi 1	neg	BR15	B2	6725 *m*/*z* and 6860 *m*/*z*
	16/01363-1 KnoLi 2	neg	BR15	B2	
	16/01363-1 KnoLi 3	neg	BR15	B2	
21	15/25396-3 KnoRe 1	O78:K80	BR32	D	not detected
	15/25396-3 KnoRe 2	O78:K80	BR32	D	
	15/25396-3 KnoRe 2	O78:K80	BR32	D	

## Data Availability

Not applicable.

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
