# Peer review of "Discrimination and Characterization of Escherichia coli Originating from Clinical Cases of Femoral Head Necrosis in Broilers by MALDI-TOF Mass Spectrometry Confirms Great Heterogeneity of Isolates"

_microorganisms, 2022, doi:10.3390/microorganisms10071472_

Round 1

Reviewer 1 Report

Nees and coauthors describe a study investigating the usage of MALDI-TOF MS for strain typing of E. coli originating from clinical cases of femoral head necrosis in broilers. In particular, the authors investigate different factors which can influence the typing by MALDI-TOF MS and its reproducibility, i.e. age of culture and culture media.

The study revealed, like others, a great diversity of of E. coli involved in FHN on protein profile level and a need of standardization with optimal conditions to perform MALDI-TOF typing.

Interestingly, the study revealed that incubation at 4°C added a peak to the spectra, this is a new finding. Is there any explanation? Could this be a modification of a protein which is apparent also in the spectra (e.g. oxidation, adduct...), mass difference to the neighboring peak might give a hint.

The authors describe a peak in B2 isolates at 9716 m/z, while other isolates depict a peak shift to 9741. Interestingly, as a new finding, they describe also six B2 isolates that show this peak shift, too. This might point on differences between human and animal derived isolates.

My most significant critic is the performance and description of methods to evaluate spectra quality for typing. I am missing objective evidence (like number of peaks, signal/noise of certain peaks, overall peak intensities,....) for the quality assessment. There is mainly subjective description of spectra after visual inspection. What are "inconsistent spectra"? This can be done better, based on their existing data set. The figures giving spectra from different cultivation times and media are not really helpful here. Maybe some zoom in would help?

Also the MSP dendrogram of the Biotyper software probably is not really applicable, as this does not work well on a strain level relationship, to my knowledge, its resolution at this level is doubtful and the method should not be used so "blind". And Fig. 4 is not helpful, unreadable anyhow.

MALDI Biotyper score limits are described as 1.7 for "genus level" and 2.0 for "species level". To my knowledge this has been changed from a certain  database version on.

Line 147-149 "Anything less than 1.7 was rated as non-identifiable by the software FlexAnalysis (version 4 Bruker Daltonics GmbH, Germany) which was used for visual inspection of the mass spectra" - this is misleading, the score 1.7 is given by the Biotyper software, FlexAnalysis is not involved. I guess the sentence has to be revised.

I do not find the long long table 1 helpful in the publication text, this might be moved to a supplement.

The citation of [23] for blood agar influence of spectra quality might not be appropriate, as this old study used a very different MALDI method which used another MALDI matrix and mass range, investigation at least partially other molecules.

Minor points:

Line 11 - broiler

Remove chapter "0"

Line 77 - shortly describe sampling (lit. 4)

Line 111 - 7560m/z

Author Response

We thank the reviewer for the time and critical comments and suggestions which helped to improve the quality of the manuscript.

Reviewer 1:

Nees and coauthors describe a study investigating the usage of MALDI-TOF MS for strain typing of E. coli originating from clinical cases of femoral head necrosis in broilers. In particular, the authors investigate different factors which can influence the typing by MALDI-TOF MS and its reproducibility, i.e. age of culture and culture media. The study revealed, like others, a great diversity of of E. coli involved in FHN on protein profile level and a need of standardization with optimal conditions to perform MALDI-TOF typing.

Interestingly, the study revealed that incubation at 4°C added a peak to the spectra, this is a new finding. Is there any explanation? Could this be a modification of a protein which is apparent also in the spectra (e.g. oxidation, adduct...), mass difference to the neighboring peak might give a hint.

We thank the reviewer for this comment. So far, we did no analysis of proteins of therefore we cannot give a conclusive answer on this. But to address this aspect we included more information in discussion, lines 314-318.

The authors describe a peak in B2 isolates at 9716 m/z, while other isolates depict a peak shift to 9741. Interestingly, as a new finding, they describe also six B2 isolates that show this peak shift, too. This might point on differences between human and animal derived isolates.

This finding is discussed in lines 352-353.

My most significant critic is the performance and description of methods to evaluate spectra quality for typing. I am missing objective evidence (like number of peaks, signal/noise of certain peaks, overall peak intensities,....) for the quality assessment. There is mainly subjective description of spectra after visual inspection.

We included now more detailed information in lines 156-179.

What are "inconsistent spectra"? This can be done better, based on their existing data set. The figures giving spectra from different cultivation times and media are not really helpful here. Maybe some zoom in would help?

We revised this accordingly, line 193. We thank the reviewer for the valid comment regarding the figure presentation. We changed Figures 1 and 2 accordingly. We now present exemplarily in Figure 1 differences in spectra intensity/quality of the 4 different media used, range 4000-600 m/z. Figure 2 exemplarily shows a decrease in intensity/quality caused by subculture: fresh culture compared to sub-cultivation 3 (20 spectra, range 4000-6000 m/z – 6000-8000, 8000-10000).

Also the MSP dendrogram of the Biotyper software probably is not really applicable, as this does not work well on a strain level relationship, to my knowledge, its resolution at this level is doubtful and the method should not be used so "blind". And Fig. 4 is not helpful, unreadable anyhow.

In several previous studies MSP dendrogram creation using the Biotyper software was applied for defining strain relationships (e.g., Alispahic et al., 2012; Alispahic et al., 2014; Lopez-Ramos et al., 2020; Lozica et al., 2020; Alispahic et al., 2022). Therefore, we would like to keep this information in our study. But we see the point of the reviewer regarding the bad resolution of Figure 4. Therefore, we included a selection of 107 strains to demonstrate the 4 clusters in better resolution.

MALDI Biotyper score limits are described as 1.7 for "genus level" and 2.0 for "species level". To my knowledge this has been changed from a certain database version on.

We are sorry for this mistake. The actual score settings are now given in lines 165-168.

Line 147-149 "Anything less than 1.7 was rated as non-identifiable by the software FlexAnalysis (version 4 Bruker Daltonics GmbH, Germany) which was used for visual inspection of the mass spectra" - this is misleading, the score 1.7 is given by the Biotyper software, FlexAnalysis is not involved. I guess the sentence has to be revised.

Again, we are sorry for this mistake and confusion. Based on a previous comment of the reviewer we revised this accordingly, lines 156-179.

I do not find the long long table 1 helpful in the publication text, this might be moved to a supplement.

We agree to the reviewers’ comment. Table 1 is now available as supplementary data.

The citation of [23] for blood agar influence of spectra quality might not be appropriate, as this old study used a very different MALDI method which used another MALDI matrix and mass range, investigation at least partially other molecules.

There is a misunderstanding, Andersen et al. (2012) is listed as reference no 24. The extraction method was performed in a very similar way as the actual recommendation of Bruker. Furthermore, the same instrument and the same matrix was used as in our study. Therefore, we will keep this citation.

Minor points:

Line 11 – broiler

Revised accordingly.

Remove chapter "0"

Removed.

Line 77 - shortly describe sampling (lit. 4)

We included a brief description of the sampling procedure, lines 80-84.

Line 111 - 7560m/z

Revised accordingly.

Reviewer 2 Report

The submitted manuscript entitled “Discrimination and characterization of Escherichia coli originating from clinical cases of femoral head necrosis in broilers by MALDI-TOF mass spectrometry confirm great heterogeneity of isolates” refers to the important problem from the poultry production as well as concerns challenging issue of the microbial identification via MALDI approach, namely, distinguishing bacteria below the species level. The size of the studied set of strains and the methodology used seem to be sufficient to draw appropriate conclusions, but I have some important suggestions that should be taken into account to increase the value of the work.

1. Firstly, a grouping of the E. coli isolates based on the MSPs dendrograms. Such an approach is available within Biotyper software and is easy to use for future users, therefore, I think it is right that the authors focused primarily on using this particular method. Nevertheless, the main goal of the work was to classify and distinguish E. coli strains/clones, which is much more complicated than simple species identification. During such analysis, we suppose to find very small differences between MS spectra, so the tool for finding such subtle changes should be sensitive and accurate enough. In my opinion, relying only on the performance of MSP is not sufficient and authors should apply other statistics to find relevant differences between strains, such as hierarchical cluster analysis (HCA), principal component analysis (PCA), or even machine learning methods (eg. artificial neural networks, ANN). A similar issue was raised by Złoch and colleagues [Złoch, M.; Pomastowski, P.; Maślak, E.; Monedeiro, F.; Buszewski, B. Study on Molecular Profiles of Staphylococcus aureus Strains: Spectrometric Approach. Molecules 2020, 25, 4894. https://doi.org/10.3390/molecules25214894] regarding S. aureus strains, where, authors used several statistical methods to find the optimal way to correct strains classification.

2. Secondly, the authors discussed the impact of the culture conditions on the spectra quality. As a possible explanation for the poor MS quality, the authors cited inhibitory effects induced by the diverse agar components, causing changes in the expression of ribosomal proteins.
Actually, the expression of the ribosomal proteins is known to be independent of the metabolic status of the cells, thus, such an explanation, in my opinion, is incorrect. A possible explanation for the observed phenomenon could be the effect of substrate components on the expression of non-ribosomal proteins, which may account for 50% of all proteins present in the bacterial extracts used for MALDI analysis. Non-ribosomal proteins, unlike ribosomal ones, are metabolic status dependent. Please revise the paragraph on this issue based on the previously mentioned work [Złoch, M.; Pomastowski, P.; Maślak, E.; Monedeiro, F.; Buszewski, B. Study on Molecular Profiles of Staphylococcus aureus Strains: Spectrometric Approach. Molecules 2020, 25, 4894. https://doi.org/10.3390/molecules25214894] and others in the field.

Author Response

We thank the reviewer for the time and critical comments and suggestions which helped to improve the quality of the manuscript.

Reviewer 2:

The submitted manuscript entitled “Discrimination and characterization of Escherichia coli originating from clinical cases of femoral head necrosis in broilers by MALDI-TOF mass spectrometry confirm great heterogeneity of isolates” refers to the important problem from the poultry production as well as concerns challenging issue of the microbial identification via MALDI approach, namely, distinguishing bacteria below the species level. The size of the studied set of strains and the methodology used seem to be sufficient to draw appropriate conclusions, but I have some important suggestions that should be taken into account to increase the value of the work.                           

  1. Firstly, a grouping of the E. coli isolates based on the MSPs dendrograms. Such an approach is available within Biotyper software and is easy to use for future users, therefore, I think it is right that the authors focused primarily on using this particular method. Nevertheless, the main goal of the work was to classify and distinguish E. coli strains/clones, which is much more complicated than simple species identification. During such analysis, we suppose to find very small differences between MS spectra, so the tool for finding such subtle changes should be sensitive and accurate enough. In my opinion, relying only on the performance of MSP is not sufficient and authors should apply other statistics to find relevant differences between strains, such as hierarchical cluster analysis (HCA), principal component analysis (PCA), or even machine learning methods (eg. artificial neural networks, ANN). A similar issue was raised by Złoch and colleagues [Złoch, M.; Pomastowski, P.; Maślak, E.; Monedeiro, F.; Buszewski, B. Study on Molecular Profiles of Staphylococcus aureus Strains: Spectrometric Approach. Molecules 2020, 25, 4894. https://doi.org/10.3390/molecules25214894] regarding S. aureus strains, where, authors used several statistical methods to find the optimal way to correct strains classification.

A very practical approach was used to investigate the feasibility of MALDI-TOF MS to characterize E. coli isolates based on a selection of well-defined isolates comprising different features. A strong interest regarding a quick identification of avian pathogenic E. coli in poultry medicine is present, and in the meantime, MALDI-TOF MS is widely used in veterinary diagnostic laboratories. Therefore, our intention was to provide data focusing on MSPs dendrogams which are mainly used for characterization of bacteria in such laboratories revealing what works and also, what does not work so far. This approach goes in line with several previous studies, e.g. Alispahic et al., 2012; Alispahic et al., 2014; Lopez-Ramos et al., 2020; Lozica et al., 2020; Alispahic et al., 2022).

But, we see the very valid comment of the reviewer and therefore, we included more information on this in discussion, lines 288-289, line 327, lines 341-343 The study of Zloch et al. (2020) was included.

  1. Secondly, the authors discussed the impact of the culture conditions on the spectra quality. As a possible explanation for the poor MS quality, the authors cited inhibitory effects induced by the diverse agar components, causing changes in the expression of ribosomal proteins.
    Actually, the expression of the ribosomal proteins is known to be independent of the metabolic status of the cells, thus, such an explanation, in my opinion, is incorrect. A possible explanation for the observed phenomenon could be the effect of substrate components on the expression of non-ribosomal proteins, which may account for 50% of all proteins present in the bacterial extracts used for MALDI analysis. Non-ribosomal proteins, unlike ribosomal ones, are metabolic status dependent. Please revise the paragraph on this issue based on the previously mentioned work [Złoch, M.; Pomastowski, P.; Maślak, E.; Monedeiro, F.; Buszewski, B. Study on Molecular Profiles of Staphylococcus aureus Strains: Spectrometric Approach. Molecules 2020, 25, 4894. https://doi.org/10.3390/molecules25214894] and others in the field.

We think there is a misunderstanding. Zloch et al. (2020) used a different instrument. The protein spectra gained by the instrument used in our study is based on ribosomal proteins. The amount of ribosomal protein expression can be affected by metabolic influences (e.g. nutrient deprivation, hypoxia, glucose starvation) – and by this result in less intensive proteomics peaks. We revised this accordingly, and thank the author for this comment. Lines 305-307.